# Impact of Anterior Malposition and Bone Cement Augmentation on the Fixation Strength of Cephalic Intramedullary Nail Head Elements

**DOI:** 10.3390/medicina58111636

**Published:** 2022-11-13

**Authors:** Torsten Pastor, Ivan Zderic, Clemens Schopper, Pascal C. Haefeli, Philipp Kastner, Firas Souleiman, Boyko Gueorguiev, Matthias Knobe

**Affiliations:** 1AO Research Institute Davos, 7270 Davos, Switzerland; 2Department of Orthopaedic and Trauma Surgery, Lucerne Cantonal Hospital, 6000 Lucerne, Switzerland; 3Department for Orthopaedics and Traumatology, Kepler University Hospital GmbH, Johannes Kepler University Linz, 4020 Linz, Austria; 4Department of Orthopaedics, Trauma and Plastic Surgery, University Hospital Leipzig, 04103 Leipzig, Germany; 5Medical Faculty, University of Zurich, 8091 Zurich, Switzerland; 6Medical Faculty, RWTH Aachen University Hospital, 52074 Aachen, Germany

**Keywords:** biomechanics, bone cement augmentation, cephalomedullary nailing, helical blade, TFNA

## Abstract

*Background and Objectives*: Intramedullary nailing of trochanteric fractures can be challenging and sometimes the clinical situation does not allow perfect implant positioning. The aim of this study was (1) to compare in human cadaveric femoral heads the biomechanical competence of two recently launched cephalic implants inserted in either an ideal (centre–centre) or less-ideal anterior off-centre position, and (2) to investigate the effect of bone cement augmentation on their fixation strength in the less-ideal position. *Materials and Methods*: Fourty-two paired human cadaveric femoral heads were assigned for pairwise implantation using either a TFNA helical blade or a TFNA screw as head element, implanted in either centre–centre or 7 mm anterior off-centre position. Next, seven paired specimens implanted in the off-centre position were augmented with bone cement. As a result, six study groups were created as follows: group 1 with a centre–centre positioned helical blade, paired with group 2 featuring a centre–centre screw, group 3 with an off-centre positioned helical blade, paired with group 4 featuring an off-centre screw, and group 5 with an off-centre positioned augmented helical blade, paired with group 6 featuring an off-centre augmented screw. All specimens were tested until failure under progressively increasing cyclic loading. *Results*: Stiffness was not significantly different among the study groups (*p* = 0.388). Varus deformation was significantly higher in group 4 versus group 6 (*p* = 0.026). Femoral head rotation was significantly higher in group 4 versus group 3 (*p* = 0.034), significantly lower in group 2 versus group 4 (*p* = 0.005), and significantly higher in group 4 versus group 6 (*p* = 0.007). Cycles to clinically relevant failure were 14,919 ± 4763 in group 1, 10,824 ± 5396 in group 2, 10,900 ± 3285 in group 3, 1382 ± 2701 in group 4, 25,811 ± 19,107 in group 5 and 17,817 ± 11,924 in group 6. Significantly higher number of cycles to failure were indicated for group 1 versus group 2 (*p* = 0.021), group 3 versus group 4 (*p* = 0.007), and in group 6 versus group 4 (*p* = 0.010). *Conclusions*: From a biomechanical perspective, proper centre–centre implant positioning in the femoral head is of utmost importance. In cases when this is not achievable in a clinical setting, a helical blade is more forgiving in the less ideal (anterior) malposition when compared to a screw, the latter revealing unacceptable low resistance to femoral head rotation and early failure. Cement augmentation of both off-centre implanted helical blade and screw head elements increases their resistance against failure; however, this effect might be redundant for helical blades and is highly unpredictable for screws.

## 1. Introduction

Trochanteric fractures cause significant socioeconomic costs and represent an increasingly common challenge for both patients and orthopaedic trauma surgeons. Individual surgeons’ skills, as well as technical aspects of the implant placement, play a crucial role for their successful fixation [1]. Although numerous advances in implant designing and postoperative treatment methods have been achieved, complication rates between 2% and 16.5% have been reported [2,3] being mostly related to cut-out, varus deformation and rotation of the femoral head fragment [4,5,6,7,8]. In recent years, novel fixation methods were developed to overcome the problematic anchoring of the implant head element (HE) in femoral heads. One of them is implemented with use of the Trochanteric Femoral Nail Advanced System (TFNA, DePuy Synthes, Zuchwil, Switzerland), allowing the choice between a helical blade or a screw HE. Other implants allow a combination of both [9]. Furthermore, bone cement may be injected through the HE into the femoral head to reduce the risk of failure in osteoporotic bone [10,11,12,13]. Beside improvements of the implants in recent years, surgeon-related technical aspects during the operation play a crucial role for patients’ outcome. The introduction of the tip–apex distance (TAD) and the calcar-related TAD already proved that off-centre positioning of the HE may predict mechanical failure of the implant [14,15,16]. However, in a clinical situation it is not always possible to achieve a perfect (centre–centre) HE position and surgeons sometimes have to accept an off-centre position of the implant [17]. Recently, a biomechanical study on artificial femoral heads demonstrated the superiority of non-augmented blades versus non-augmented screws in an off-centre position. Furthermore, bone cement augmentation was able to enhance the anchorage of off-centre-positioned HE to a level of centrally placed cephalic implants [18]. However, the resistance to failure of malpositioned non-augmented helical blades and screw head elements, as well as the effect of bone cement injection on a malpositioned implant, have not been investigated in cadaveric bone yet. Therefore, the aims of this study were to investigate in human cadaveric femoral heads (1) the biomechanical competence of two recently launched cephalic implants inserted in either ideal (centre–centre) or less-ideal anterior off-centre positions and (2) to investigate the effect of bone cement augmentation of the cephalic implants on their fixation strength in a less ideal position.

## 2. Materials and Methods

### 2.1. Specimens and Study Groups

Forty-two fresh frozen (−20 °C) paired human cadaveric femoral heads from 10 females and 11 males, aged 68.3 years on average (range 54–82 years), were used. All donors gave their informed consent inherent within the donation of the anatomical gift statement during their lifetime (Science Care, Inc., Phoenix, AZ, USA). All specimens underwent high-resolution peripheral quantitative computed tomography (HR-pQCT, Xtreme CT, SCANCO Medical AG, Brüttisellen, Switzerland) to exclude any bone pathologies and calculate volumetric bone mineral density (BMD) within a cylinder of 20 mm diameter and 30 mm length, located in the centre of the femoral head, using a phantom (European Forearm Phantom QRM-BDC/6, QRM GmbH, Möhrendorf, Germany). The specimens were assigned for pairwise implantation using either a TFNA helical blade or a TFNA screw HE. The HEs of each type (helical blade or screw) were implanted in either centre–centre or 7 mm anterior off-centre position. Next, 7 paired specimens implanted with helical blades and screws in the anterior off-centre position were augmented with bone cement (Traumacem V+, DePuy Synthes, Zuchwil, Switzerland). Thus, six study groups were created, consisting of 7 specimens each and combined in 3 clusters, comprising specimens of the same donors in both paired groups of each cluster: group 1 with a centre–centre-positioned helical blade, paired with group 2 featuring a centre–centre screw (cluster 1); group 3 with an off-centre-positioned helical blade, paired with group 4 featuring an off-centre screw (cluster 2); and group 5 with an off-centre-positioned augmented helical blade, paired with group 6 featuring an off-centre augmented screw (cluster 3, Figure 1 and Figure 2). The sample size of 7 specimens per group was considered sufficient for detection of existing significant differences among the corresponding groups, based on previous published work with similar study design, investigating different fixation methods in femoral heads [19,20,21].

### 2.2. Specimens Preparation

All femoral heads were sawed 50 mm distally to the articular surface and orthogonally to the femoral neck axis after thawing for 24 h at room temperature. Implantation was performed according to the manufacturer’s guidelines under fluoroscopic control (Siemens ARCADIS Varic, Siemens Medical Solutions AG, Erlangen, Germany) with a targeted TAD of 20 mm [14]. According to the group assignment, a guide wire was either placed centrally or with a 7 mm anterior offset at a depth of 40 mm into the femoral head perpendicular to the cut surface, and therefore parallel to the femoral neck axis. For this purpose, the cutting plane of each femoral head was divided into four quadrants defined by distance measurements (Figure 1). For off-centre implant insertion, the entry point was located 7 mm anteriorly to the centre of the femoral head. The 7 mm anterior off-centre position was in agreement with previous work on cephalic implant positioning and seems to reflect well the reality in the surgical theatre [22]. All HEs had a length of 100 mm. The helical blades were inserted over the guide wire to their final depth using hammer blows without predrilling. The screws were implanted after predrilling with a 6 mm drill bit to the desired depth. They were inserted over the guide wire and tightened. Both helical blade and screw HEs were orientated as in a real patient in order to fit within the locking mechanism of the nail. Femoral heads assigned for bone cement augmentation were warmed up to 37 °C in a water bath (Y6, Grant Instruments Cambridge Ltd., Shepreth, UK) prior to bone cement injection. A total volume of 3 mL bone cement was injected into the specimens in a standardized manner under fluoroscopic control. After injection of 1 mL through the HE’s perforations on the cranial side, the canula was twisted 180° and another 1 mL was injected through the caudal perforations of the HEs. Next, the cannula was withdrawn 10 mm and the procedure was repeated with injection of 0.5 mL twice [18]. All specimens underwent CT examination to exclude possible undesired bone damages created during implantation.

### 2.3. Test Setup

Biomechanical testing was performed on a servo-hydraulic test system (Acumen III, MTS Systems Corp., Eden Prairie, MN, USA) equipped with a 3 kN load cell in a dry environment at room temperature (20 °C). The test setup was adopted from previous studies and simulated an unstable trochanteric fracture with lack of medial support and load sharing at the fracture gap (Figure 3) [18,23,24]. To mimic the locking mechanism of the TFNA nail that allows sliding without rotation of the HEs, the implant shafts were inserted in flange sleeves. These were rigidly mounted on a base fixture with a total inclination of 149° to the vertical line to simulate a 130° caput-collum-diaphyseal angle, a 16° resultant joint load vector orientation to the vertical, and 3° lateral inclination of the femoral shaft axis as previously described [24]. The implants were free to slide along their shaft axis with blocked rotation around it during testing. The femoral heads were attached to spikes on a polycarbonate plate mounted on a roller bearing, allowing for rotational movement of the plate and the femoral head around its axis. Furthermore, the specimens were mounted on two cylindrical rollers allowing varus and valgus tilting. Axial load was transmitted to the femoral heads via a polymethylmethacrylate (PMMA) shell mounted on a XY-table to compensate for shear moments during cyclic testing. Furthermore, reflective markers were attached to the femoral head and the HE for optical motion tracking.

### 2.4. Loading Protocol

Progressively increasing cyclic axial loading at 2 Hz, starting at 1000 N and being with physiologic profile of each cycle, was applied until failure [25]. The peak load of each cycle increased monotonically by 0.1 N/cycle until reaching 3000 N, while its valley load was kept constant at 100 N. If the specimens reached 3000 N without failure, the test was continued with no further increase of the peak load. Test stop criterium was reaching a 10 mm axial displacement of the machine actuator relative to the test start.

### 2.5. Data Acquisition and Analysis

Machine data in terms of axial load and axial displacement were recorded from the machine controllers at 128 Hz. Based on these data, initial axial construct stiffness was calculated from the ascending slope of the load–displacement curve between 400 N and 600 N during the first loading cycle. Anteroposterior X-rays were taken every 500 cycles using a triggered C-arm. Furthermore, the coordinates of the optical markers attached to the femoral head and the HE were continuously acquired throughout the tests at 75 Hz by means of stereographic optical measurements using contactless full-field deformation technology (Aramis SRX, GOM GmbH, Braunschweig, Germany) to assess the bone-implant motions in all six degrees of freedom with regard to the marker sets. Anatomical axes (vertical, frontal and sagittal) of the femoral head and the HE axis were defined by proper alignment of the respective marker sets. Varus deformation was defined as the relative bending of the femoral head to the HE axis in the coronal plane. Furthermore, rotation of the femoral head around the HE axis was evaluated. Implant cut-out and implant migration (cut-through) were defined as relative cranial movement of the HE in the femoral head and relative longitudinal HE movement along its axis, respectively. The outcome values of these four parameters were analyzed after 2000 and 4000 cycles, and if applicable after 6000, 8000 and 10,000 cycles in peak loading condition, to evaluate the degradation of the construct stability over the course of cycles [19]. Margins of 5° varus deformation and 10° rotation of the femoral head around the implant axis—considered with respect to the beginning of the cyclic test—were defined as clinically relevant failure criteria derived from previous work [10,23,26]. For each separate specimen, the numbers of cycles until fulfilment of each of these two criteria under peak loading condition were calculated. Based on these, clinical failure was defined as the event when whichever of the two criteria was fulfilled first, and the corresponding number of cycles until that event was considered as cycles to clinical failure. Catastrophic failure modes were evaluated using X-ray imaging and physical inspection of the implant in end of each test.

### 2.6. Statistical Analysis

Statistical analysis was performed with SPSS software package (IBM SPSS Statistics, V27, IBM, Armonk, NY, USA). Shapiro–Wilk test was conducted to prove normality of data distribution for each separate parameter and group. Explorative data was calculated in terms of mean value and standard deviation (SD). For the single-measure parameters BMD, initial stiffness and cycles to clinical failure, significant differences between the paired groups—belonging to the same cluster—were explored with Paired-Samples t-tests. Furthermore, One-Way Analysis of Variance (ANOVA) was conducted to screen these parameters for significant differences with regard to the other pairs of groups associated with the same implanted HE (blade or screw), but assigned to a different cluster (e.g., all groups featuring TFNA blade implantation were compared amongst each other with regard to centre–centre, off-centre, and augmented off-centre positioning). For the longitudinal multiple-measure parameters cut-out, implant migration, rotation around implant axis and varus deformation at the pre-defined time points of cyclic testing, outcome measures among all groups were screened for significant differences with General Linear Model (GLM) Repeated Measures (RM) tests. Thereby, the number of repeated-measures steps was determined under consideration of the highest rounded number of predefined cycles when none of the specimens within the compared groups had failed yet. If any of the ANOVA or GLM RM tests indicated overall significance, a post hoc test analysis accounting for multiple comparisons was conducted. Significance level was set to 0.05 for all statistical tests.

## 3. Results

### 3.1. Morphometrics

Mean age of the donors was 69.4 ± 4.9 years in groups 1 and 2, 74.2 ± 3.5 years in groups 3 and 4, and 66.0 ± 8.6 years in groups 5 and 6, with no significant differences among all groups, *p* = 0.121. BMD (mgHA/cm^3^) was 186.5 ± 36.6 in group 1, 180.3 ± 51.8 in group 2, 183.2 ± 37.6 in group 3, 176.6 ± 35.4 in group 4, 180.6 ± 45.3 in group 5 and 179.5 ± 65.9 in group 6, with no significant differences among all groups (*p* = 0.999).

### 3.2. Initial Stiffness

Initial axial stiffness (N/mm) was 1211.1 ± 85.6 in group 1, 1168.2 ± 260.6 in group 2, 1471.8 ± 553.8 in group 3, 973.7 ± 331.1 in group 4, 1169.6 ± 433.6 in group 5 and 1214.7 ± 309.8 in group 6. No significant differences were detected within each cluster (*p* ≥ 0.104) as well as among all groups (*p* = 0.388).

### 3.3. Varus Deformation, Femoral Head Rotation, Implant Migration and Implant Cut-Out at Predefined Cycles

The outcome measures for these four parameters of interest are summarized in Table 1.

In the centre–centre position and augmented off-centre position, there were no significant differences detected between the two HE designs in the paired groups 1–2 (cluster 1) and 5–6 (cluster 3, *p* ≥ 0.077), respectively. However, in the non-augmented off-centre position, the screw HEs in group 4 were associated with significantly higher values compared to the helical blade HEs in the paired group 3 for rotation around the implant axis and cut-out (*p* ≤ 0.047), with a trend toward significantly higher values for varus deformation (*p* = 0.052), and with non-significantly higher values for implant migration (*p* = 0.122). Furthermore, the off-centre screw positioning in group 4 was associated with significantly higher values compared to the centre–centre screw positioning in group 2 for rotation around the implant axis and cut-out (*p* ≤ 0.008). No significant differences between the corresponding groups with helical blade implantation (groups 3 and 1) were detected (*p* ≥ 0.579). On the other hand, whereas the augmentation of off-centre screws in group 6 resulted in significantly lower values compared to group 4 with non-augmented off-centre screws for varus deformation, rotation around implant axis, and cut-out (*p* ≤ 0.026), the differences between the corresponding groups 3 and 5 with helical blade implantation were not significant (*p* ≥ 0.227).

### 3.4. Cycles to Clinical Failure

Cycles to clinical failure (5° varus or 10° rotation of the femoral head, whichever occurred first) were 14,919 ± 4763 in group 1, 10,824 ± 5396 in group 2, 10,900 ± 3285 in group 3, 1382 ± 2701 in group 4, 25,811 ± 19,107 in group 5 and 17,817 ± 11,924 in group 6 (Figure 4). Centre–centre positioning in cluster 1 resulted in significantly higher resistance to failure in group 1 versus group 2 (*p* = 0.021). Moreover, augmented off-centre positioning in cluster 3 resulted in no significant difference between the paired groups with blade and screw implantation (*p* = 0.193). However, non-augmented off-centre HE positioning in cluster 2 was associated with a significantly higher number of cycles to failure in group 3 using helical blades versus group 4 using screws (*p* = 0.007). Finally, augmented off-centre screw positioning in group 6 resulted in significantly higher number of cycles to failure compared to non-augmented screw positioning in group 4 (*p* = 0.010). No further significant differences were detected among all other non-paired groups (*p* ≥ 0.112).

### 3.5. Catastrophic Failure Modes

Whereas centre–centre screw positioning resulted in two failure cases by rotation around the implant axis and five failure cases by varus collapse, non-augmented and augmented off-centre screw positioning was associated with exclusive rotational failure around the HE axis. On the other hand, rotational failure in the groups with blade implantation was detected in three specimens with centre–centre positioning, four specimens with non-augmented and four specimens with augmented HE off-centre positioning.

## 4. Discussion

Trochanteric fractures are a significant burden for health systems as most patients need to be hospitalized and operated [1]. When TFNA is used to treat those fractures, surgeons have the choice to select intraoperatively either a helical blade or a screw as a HE for fixation of the femoral head and neck. Furthermore, it offers the option for bone cement augmentation. The current study investigated the biomechanical characteristics of these two different HEs in the ideal centre–centre and less-ideal anterior off-centre positions. Moreover, the effect of bone cement augmentation on the fixation strength within the femoral head was explored.

Comparable initial construct stiffness independent from the implant positioning or augmentation with bone cement was reported in the current study. Furthermore, similar results were observed in optimally positioned implants, although helical blades demonstrated a slightly better resistance to varus deformation when compared to screws. These findings are in line with other reports in the literature, demonstrating a trend of higher cut-out rates when using screws versus helical blades [10,27]. Furthermore, the current study revealed a significantly higher resistance of uncemented helical blades to rotational forces and moments in the centre–centre position compared to centrally positioned uncemented screws. A possible explanation for this might be the design of the helical blade, which compacts the cancellous bone in the femoral head during insertion [28]. This theoretically provides better fixation strength in low bone quality by preventing bone loss, because pre-drilling along the entire HE length—as required for use of cephalic screws in adequate bone quality—is not always necessary [29]. On the other hand, an increased resistance to the rare complication related to medial cut-through of the HE along its axis, as well as higher pull-out forces, were reported for cephalic screws [30,31]. However, in contrast to other reports in the literature, the current study revealed no significant differences regarding HE migration along its axis among all investigated groups. Furthermore, there are several existing reports in favor of centrally-placed screw HEs compared to helical blades [30,32]. In the current study, the bone compaction around the helical blade might also be an explanation for the higher resistance to rotational moments following anterior malpositioning when compared to screws. Further, the helical blades in the anterior off-centre position were significantly less susceptible for failure and compensated the offset significantly more effective than the screws. This is in line with previous results reported by Sermon et al., who investigated anterior malpositioned implants in osteoporotic foam models [18]. In addition, they investigated malpositioned helical blades and screws with anterior and posterior offset and reported no differences between them. It is therefore hypothesized that although only anteriorly malpositioned implants were investigated in the current study, the results are transferable to the posterior malposition, too. However, despite the higher resistance to failure of the malpositioned helical blades compared to screws, this study fully supports the well-established mantra that correct implant placement in the centre–centre position is of utmost importance.

Various reports in the literature demonstrate an increase in resistance to failure of helical blades and screws augmented with bone cement [10,19,23,33,34,35,36]. Furthermore, a recently published review reported fewer reoperations, less complications and shorter hospital stay at the cost of a slightly increased operation time when bone cement was used for augmentation in elderly patients [37]. In contrast, no advantages in resistance to both failure load and axial displacement after cement augmentation of intramedullary nailed trochanteric fractures was reported by Fensky et al. [38]. Moreover, cement augmentation of cannulated screws in a femoral neck fracture model did not demonstrate any improvement in construct stability [39].

However, most of these studies focused on an optimally positioned HE, with only one of them focusing on cement augmentation of a malpositioned implant [18]. The findings of the current study also demonstrate an increased resistance to failure after cement augmentation of both investigated HEs in off-centre position, although this effect was only significant for screws, demonstrating unacceptably low resistance to failure following off-centre screw positioning without cement augmentation in the currently used pool of specimens. In consequence, the findings of the present study suggest that the anterior off-centre position must be avoided for screws in a real patient under all circumstances. If the guide wire of the TFNA system is not revisable in an anterior or posterior malposition, a screw should be avoided, and a blade should be inserted instead. If a screw is already inserted in an off-centre position and is not revisable, the results of the current study suggest its augmentation with bone cement. However, the data scattering in the current study might be an indicator of an unpredictable outcome of this approach in real patients, which should be taken into account by a very careful aftercare of patients with a malpositioned augmented screw. In addition, bone cement augmentation of a non-revisable helical blade in an anterior off-centre position might not always be beneficial. Although in the current investigation their resistance to failure was higher when compared to non-augmented off-centre helical blades, the scattering of the data prohibited significance, and therefore the downsides of bone cement augmentation should be carefully balanced against its possible advantages in a clinical situation.

Another point worth mentioning is that during the biomechanical analysis this study focused on clinically relevant findings. For this reason, no comparisons were made between non-augmented off-centre-positioned screws (group 4) and augmented off-centre-positioned helical blades (group 5), despite the expected significant difference between them (see Figure 4). In a clinical setting, a misplaced screw would be unlikely to be removed and replaced with an augmented off-centre helical blade—due to the necessary predrilling for screw HEs, bone impaction during insertion of the helical blades cannot occur.

This study has some limitations inherent to all biomechanical investigations. First, only a limited number of femoral heads were used per group, restricting the generalization of the study findings; however, an appropriate paired study design was set to compare the biomechanical competence of the two different implants. Moreover, a bone model is not capable to completely simulate in vivo situations with swelling and biological reactions of the surrounding soft tissues following a bone fracture in a real human. Furthermore, the applied biomechanical model did not consider all in vivo forces and moments acting on the femoral head; however, the test setup and loading protocol were defined in such a way to ensure a close simulation of dynamic physiologic loading conditions. Due to the paired study design, only screws and helical blades inserted in the same position could be investigated in the same donor; however, prior to testing, a BMD evaluation demonstrated equally distributed values, thus ensuring comparability among the different groups. Other limitations are artificially created fractures via osteotomies, which do not necessarily obey the physical laws of real fracture mechanisms; however, they were used for standardization purposes and better implant comparability. Despite these limitations, the main failure modes in the current study reflected well clinical failure types in real patients—rotation and varus tilting of the femoral head [40]. However, large scattering of the data was observed in both bone-cement-augmented groups; therefore, further studies are needed to investigate the biomechanical behavior of malpositioned cephalic implants and the influence of different cement distribution models, especially in osteoporotic bone. Moreover, several factors determine the clinical outcome besides the implant design, such as duration of surgery, consequences following cement augmentation, quality of reduction, soft tissue damage, infections, and other postoperative complications. Further prospective randomized clinical trials are needed to relate the findings of the current study to the clinical practice.

## 5. Conclusions

From a biomechanical perspective, proper centre–centre implant positioning in the femoral head is of utmost importance. In cases when this is not achievable in a clinical setting, a helical blade is more forgiving in the less ideal (anterior) malposition when compared to a screw, the latter revealing unacceptably low resistance to femoral head rotation and early failure. Cement augmentation of both off-centre implanted helical blades and screw head elements increases their resistance against failure; however, this effect might be redundant for helical blades and is highly unpredictable for screws.

## Figures and Tables

**Figure 1 medicina-58-01636-f001:**
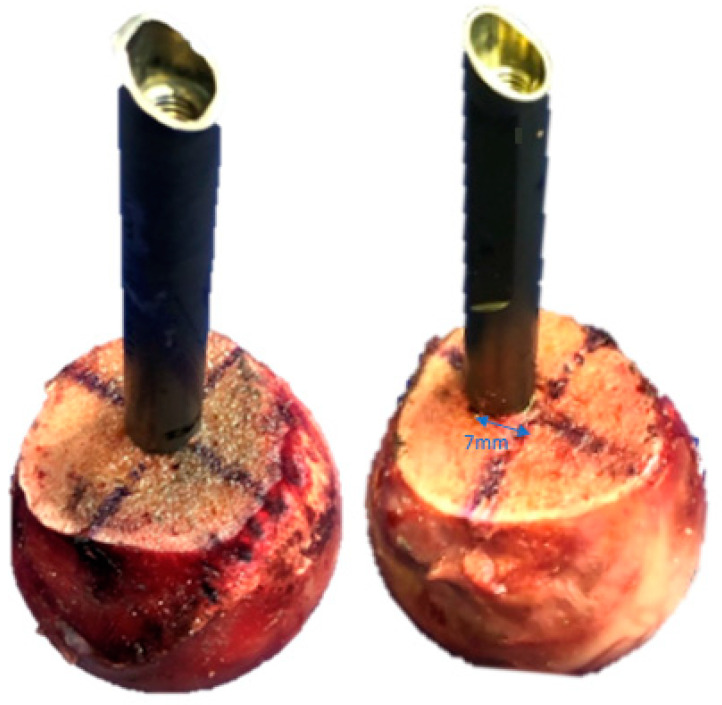
Exemplified specimens representing the implantation in groups 1 and 2 (**left**) and groups 3–6 (**right**). Note the 7 mm anterior off-centre position of the HEs in groups 3–6.

**Figure 2 medicina-58-01636-f002:**
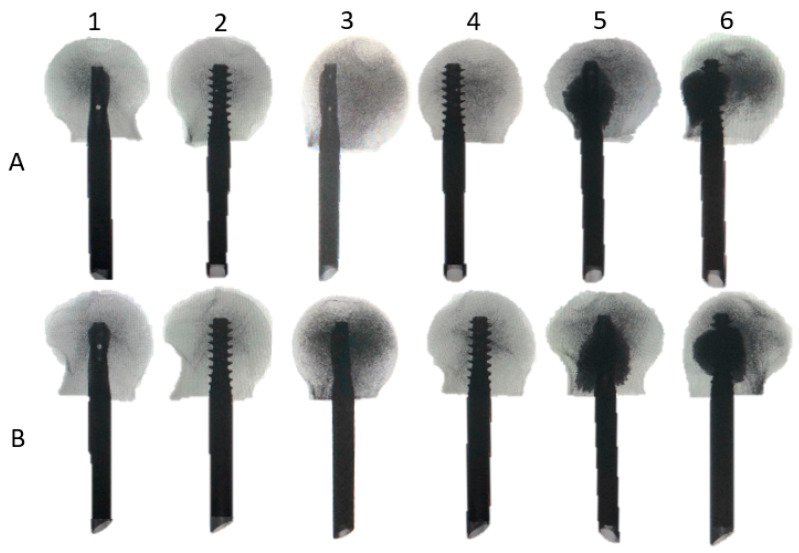
Exemplified samples of each group in superoinferior (**A**) and anteroposterior (**B**) views; groups 1 and 2: helical blade and screw in centre–centre position; groups 3 and 4: helical blade and screw in 7 mm anterior off-centre position; groups 5 and 6: helical blade and screw in 7 mm anterior off-centre position augmented with bone cement.

**Figure 3 medicina-58-01636-f003:**
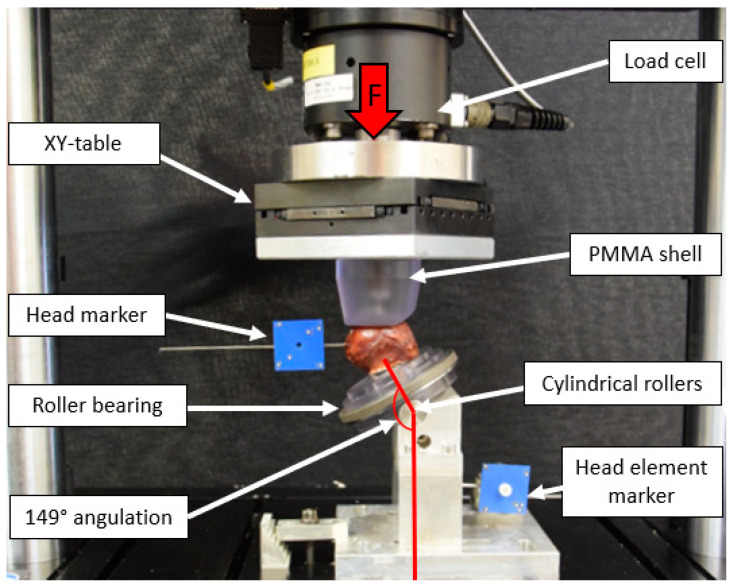
Setup with a specimen mounted in 149° implant shaft inclination to the vertical line for biomechanical testing. Vertical arrow (F) denotes loading direction.

**Figure 4 medicina-58-01636-f004:**
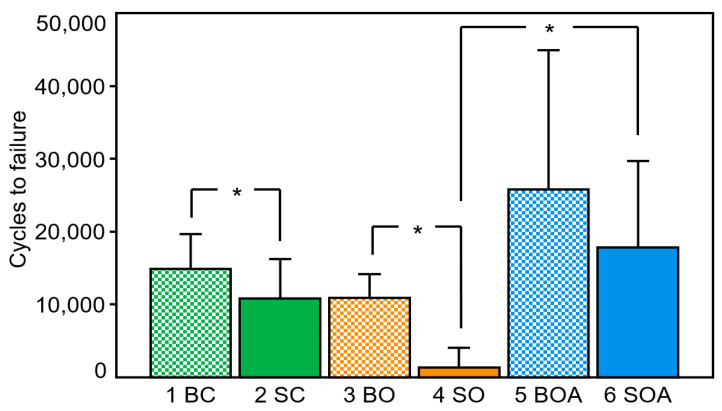
Cycles to clinical failure presented for each separate group in terms of mean and SD. Clusters: 1 (green); 2 (orange); 3 (blue); helical blades: checkerboard pattern; screws: solidly filled. BC: group 1 (helical blade centre–centre); SC: group 2 (screw centre–centre); BO: group 3 (helical blade off-centre); SO: group 4 (screw off-centre); BOA: group 5 (helical blade off-centre augmented); SOA: group 6 (screw off-centre augmented). Stars indicate significant differences.

**Table 1 medicina-58-01636-t001:** Outcome measures for the investigated longitudinal multi-measure parameters of interest varus deformation, femoral head rotation, implant migration and cut-out, presented separately for each study group at the predefined numbers of cycles in terms of mean and SD. The six groups were combined in three clusters comprising specimens of the same donors each—group 1 (centre–centre positioned blade) paired with group 2 (centre–centre positioned screw), group 3 (off-centre positioned blade) paired with group 4 (off-centre positioned screw), and group 5 (off-centre positioned augmented blade) paired with group 6 (off-centre positioned augmented screw).

Parameter	Cycles	Study Groups					
		Cluster 1		Cluster 2		Cluster 3	
		1 Blade Centre– Centre	2 Screw Centre– Centre	3 Blade Off-Centre	4 Screw Off-Centre	5 Blade Off-Centre Augmented	6 Screw Off-Centre Augmented
Varus deformation [deg]	2000	1.75 ± 0.67	2.17 ± 0.90	1.20 ± 1.02	3.33 ± 2.07	1.06 ± 0.40	1.47 ± 0.51
	4000	2.16 ± 0.82	2.91 ± 1.49	1.61 ± 1.45	5.15 ± 2.58	1.20 ± 0.50	1.74 ± 0.69
	6000	2.53 ± 0.90	3.77 ± 2.07	2.25 ± 1.88	–	1.37 ± 0.65	2.09 ± 1.00
	8000	3.01 ± 1.05	4.89 ± 2.62	3.12 ± 2.37	*–*	1.51 ± 0.85	*–*
	10,000	3.44 ± 1.55	6.08 ± 3.07	5.24 ± 6.29	*–*	1.69 ± 1.05	*–*
Femoral head rotation [deg]	2000	1.03 ± 1.90	0.84 ± 1.74	0.68 ± 0.42	25.60 ± 17.02	0.54 ± 0.31	1.60 ± 1.67
	4000	1.32 ± 1.93	2.24 ± 5.23	1.35 ± 0.76	30.97 ± 25.50	0.66 ± 0.39	2.92 ± 3.50
	6000	1.65 ± 2.09	3.73 ± 7.71	2.73 ± 1.45	*–*	0.84 ± 0.51	5.82 ± 7.81
	8000	2.55 ± 3.68	6.20 ± 9.79	5.88 ± 3.42	*–*	1.22 ± 0.76	*–*
	10,000	5.03 ± 7.78	8.32 ± 11.44	10.74 ± 10.77	*–*	2.02 ± 1.55	*–*
Implant migration [mm]	2000	0.09 ± 0.11	0.16 ± 0.15	0.60 ± 0.02	0.40 ± 0.30	0.06 ± 0.06	0.07 ± 0.02
	4000	0.20 ± 0.31	0.24 ± 0.20	0.10 ± 0.03	0.37 ± 0.39	0.08 ± 0.06	0.09 ± 0.04
	6000	0.31 ± 0.42	0.31 ± 0.25	0.30 ± 0.27	*–*	0.11 ± 0.07	0.14 ± 0.08
	8000	0.64 ± 0.89	0.48 ± 0.31	0.59 ± 0.42	*–*	0.19 ± 0.11	*–*
	10,000	1.42 ± 2.20	0.96 ± 0.67	0.93 ± 0.49	*–*	0.37 ± 0.29	*–*
Implant cut-out [mm]	2000	1.09 ± 0.35	1.23 ± 0.23	1.11 ± 0.53	3.53 ± 1.75	1.02 ± 0.19	1.28 ± 0.38
	4000	1.38 ± 0.44	1.55 ± 0.42	1.38 ± 0.75	4.28 ± 2.44	1.18 ± 0.25	1.51 ± 0.49
	6000	1.63 ± 0.53	1.98 ± 0.67	1.70 ± 1.02	*–*	1.35 ± 0.35	1.76 ± 0.64
	8000	1.90 ± 0.67	2.61 ± 1.07	2.16 ± 1.32	*–*	1.54 ± 0.43	*–*
	10,000	2.30 ± 1.12	3.53 ± 2.08	4.33 ± 5.45	*–*	1.74 ± 0.53	*–*

## Data Availability

The datasets used and/or analyzed during the current study are available from the corresponding author on reasonable request.

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
