# Peer review of "Impact of Anterior Malposition and Bone Cement Augmentation on the Fixation Strength of Cephalic Intramedullary Nail Head Elements"

_medicina, 2022, doi:10.3390/medicina58111636_

Round 1

Reviewer 1 Report

First of all thank you for this biomechanical study. My overall impression of your conducted study of different fixation strategies of femoral heads is very well designed and the presentation of your results are well strucutred.

You can find my questions and comments below. 

Questions:

- Were pre-damages created by the implantation excluded by CT scans prior to the actual tests?

- The measurement of bone marrow density showed quiet low values which might come from osteoporosiss taking the age of 69,4 into account. In the discussion you mention that you are comparing your results to osteoporotic bones (line 321 and 385, introduction line 67), which absolutely makes sense. But you never mentioned before that your specimen are osteoporotic, too (my assumption). Maybe add this in your abstract and/or M&M part by using the word osteoporotic directly and not just indicating BMD numbers.

Corrections:

- Is there any explanation for the reduction of femoral head rotation for group “blade center center” from cycle 8000- 10.000? Is it a global change that is given here or do the numbers add on to the previous? From my understanding of the experimental design and description in your text it is the absolute number not adding up.

- It would be nice if it is somehow possible to include the most important significant changes into the table. Maybe with something like a “significance legend” or mentioning the number to which it is significant by indicating it as high-positioned numbers (e.g. result of group 1 = 1.75 ±0.67 3,5,XYZ) Just if it does not destroy the whole table format.

- Y missing in line 224 74.2+-3.5 (y)ears -> line 225

- Does it need the “(yes/no)” thing in line 301?

- Line 358: Seems like there is one space between “another” and “ worth” too much.

Best regards

Reviewer

Author Response

See authors' reply in the attached file.

Reviewer 2 Report

This study revealed centre placement and helical blade  were biomechanically superior to off-centre placement and screw, using frozen femoral heads and  test machine. The indication of TFNA is mainly to intertrochanteric fractures,  however, this results were significant and relevant to real surgery.

Correction;

L104 featurin-----featuring

Author Response

(The authors gave the same response as above.)
